# Process evaluation of a randomised controlled trial aimed at improving health behaviours and vitamin D status during pregnancy: Implementation of the SPRING trial

Simone Proebstl[1,2,3], Christina Vogel[3,4,5,6], Wendy Lawrence[7], Sofia Strömmer[3], Hazel Inskip[3,4], Julia Hammond[3], Kate Hart[3], Karen McGill[3], Nicholas C. Harvey[3,4], Mary Barker[3,4,8‡], Janis Baird[3,4‡*]

1 Institute for Medical Information Processing, Biometry, and Epidemiology - IBE, LMU Munich, Munich, Germany, 2 Pettenkofer School of Public Health, Munich, Germany, 3 MRC Lifecourse Epidemiology Centre, University of Southampton, Southampton, United Kingdom, 4 NIHR Southampton Biomedical Research Centre, University of Southampton and University Hospital Southampton NHS Foundation Trust, Southampton, United Kingdom, 5 Centre for Food Policy, University of London, London, United Kingdom, 6 NIHR Applied Research Collaboration Wessex, Southampton, United Kingdom, 7 Faculty of Medicine, University of Southampton, Southampton, United Kingdom, 8 Faculty of Environmental and Life Sciences, School of Health Sciences, University of Southampton, Southampton, United Kingdom

‡ These authors are joint senior authors on this work.
* jb@mrc.soton.ac.uk

## Abstract

### Background

The Southampton PRegnancy INtervention for the Next Generation (SPRING) aimed to assess the efficacy of vitamin D supplementation and the behaviour change intervention 'Healthy Conversation Skills' (HCS) in improving the nutritional status of pregnant women. This paper describes the implementation of these interventions. Efficacy of HCS in improving diet quality and physical activity was evaluated in subgroups of women who discussed ways to improve these behaviours.

### Methods

In total, 717 pregnant women were recruited from a maternity hospital in Southampton, England. Quantitative data were collected using questionnaires, case report forms, and audio recordings. Following Medical Research Council guidance, fidelity, dose, and reach were evaluated descriptively. Multiple linear regression models were produced for subgroup analyses.

### Results

Research nurses demonstrated high competence in using HCS. Compliance with intervention protocols for delivering and receiving both interventions was high. Participants took a median of 96% of the supplements and most women (85%) attended

**Data availability statement:** The data will be uploaded to an open access data repository at the University of Southampton https://pure.soton.ac.uk/.

**Funding:** This work was supported by the UK Medical Research Council [MC_PC_21003; MC_PC_21001] and National Institute for Health and Care Research National Institute for Health Research (NIHR) Southampton Biomedical Research Centre, University of Southampton and University Hospital Southampton NHS Foundation Trust. The work leading to these results was supported by the European Union's Seventh Framework Programme (FP7/2007–2013), projects EarlyNutrition, ODIN and LifeCycle under grant agreements numbers 289346, 613977 and 733206, and by the BBSRC (HDHL-Biomarkers, BB/P028179/1 and BB/P028187/1), as part of the ALPHABET project, supported by an award made through the ERA-Net on Biomarkers for Nutrition and Health (ERA HDHL), Horizon 2020 grant agreement number 696295. We are extremely grateful to Merck GmbH for the kind provision of the Vigantoletten supplement. The funders had no role in study design, data collection and analysis, decision to publish, or preparation of the manuscript. There was no additional external funding received for this study. The funders (UK MRC, NIHR Southampton BRC, University of Southampton and University Hospital Southampton NHS Foundation Trust) provided support in the form of salaries for authors [NCH, JB, MEB, CC, MB, WTL, CV, SS, HMI, JH, KM, KH] but did not have any additional role in the study design, data collection and analysis, decision to publish, or preparation of the manuscript. The specific roles of these authors are articulated in the 'author contributions' section. For the purpose of Open Access, the author has applied a Creative Commons Attribution (CC BY) licence to any Author Accepted Manuscript version arising from this submission.

**Competing interests:** Janis Baird, Mary Barker and Wendy Lawrence have received grant research support from Danone Nutricia Early Nutrition. Cyrus Cooper has received

all four Healthy Conversations sessions. Women of lower socioeconomic status and from ethnic minorities were under-represented amongst participants. Findings were not sufficient to suggest an effect of HCS on diet quality among those who discussed diet but indicated a marginally beneficial effect on physical activity among those who discussed physical activity. Results suggested a weak dose-dependent effect, with the most pronounced difference in physical activity between the control group and the intervention sub-group with the highest exposure (adjusted difference 0.16 SD (95%-CI −0.03; 0.34)).

## Conclusion

This process evaluation confirms that the intervention components were delivered with high fidelity and rates of compliance. Altering dietary behaviours proved more challenging than altering physical activity behaviours. Research is needed to explore barriers to healthy eating faced by women during pregnancy and how these can be overcome. This paper also highlights the difficulty of engaging people from ethnic minorities and disadvantaged backgrounds in research.

## Introduction

In England, about half of women experience overweight or obesity during pregnancy [1]. Maternal obesity and gestational weight gain beyond recommendations increase the risk of pregnancy complications such as gestational diabetes and hypertensive disorders [2–4]. In addition, maternal adiposity has negative consequences for the health of the offspring. Early life exposures such as maternal diet and body composition influence the growth and development of the fetus, which in turn affects the risk of that child developing non-communicable diseases later in life [5]. Children of mothers with obesity are at a higher risk of being overweight or obese themselves [6] and suffering from diabetes [7] and cardiovascular disease [8]. Hence, improving health behaviours during pregnancy not only enhances women's own health but also holds potential to improve their children's health in the future.

Pregnancy is considered a unique opportunity to implement behaviour change interventions. Women experience higher levels of motivation because they are concerned about their baby's health and are in regular contact with the healthcare system [9]. However, pregnancy might also introduce new barriers to altering or sustaining behaviour change [10,11] and trials often fail to show effects that exceed small or short-term improvements [12,13]. Therefore, further research is needed to improve understanding of the factors that can impact the effectiveness of lifestyle interventions during pregnancy.

The Southampton PRegnancy Intervention for the Next Generation (SPRING) trial [14] aimed to investigate the efficacy of two different approaches to improving the health status of women during pregnancy – a behaviour change intervention and micronutrient supplementation. SPRING combines the interventions from the Southampton Initiative for Health (SIH) [15] and the Maternal Vitamin D Osteoporosis Study (MAVIDOS) [16]. SIH investigated the effectiveness of the behaviour change

consultancy, lecture fees and honoraria from AMGEN, GSK, Alliance for Better Bone Health, MSD, Eli Lilly, Pfizer, Novartis, Servier, Merck, Medtronic and Roche. Members of Hazel Inskip's team have received grant research support from Nestec and Danone Nutricia Early Life Nutrition. Nicholas Harvey has received consultancy, lecture fees and honoraria from Alliance for Better Bone Health, AMGEN, MSD, Eli Lilly, Servier, Shire, UCB, Consilient Healthcare, Theramex, Kyowa Kirin and Internis Pharma. The commercial companies that Professors Harvey and Cooper undertook consultancy work for did not play any role in this study. Professors Havey and Cooper were not directly employed by these companies. None of the interests declared alter our adherence to PLOS ONE policies on sharing data and materials.

intervention Healthy Conversation Skills (HCS) in improving women's diet and physical activity, while MAVIDOS assessed the effect of maternal vitamin D supplementation on neonatal bone mineral content. An insufficient vitamin D status during pregnancy is common [17] and linked to both negative maternal and child health outcomes [18]. Hence, antenatal vitamin D supplementation could be an effective public health measure to improve maternal and child health [19].

To understand the complexity of the outcome results from the SPRING trial, a thorough process evaluation is crucial, particularly for the behaviour change component. These insights will help understanding of whether trial outcomes could be reproduced and how the intervention could be replicated in different contexts [20]. The results of the outcome evaluation will be published separately and will be informed by the findings of this process evaluation. According to the Medical Research Council (MRC) Guidance on Complex Interventions, process evaluation should be used to assess fidelity and quality of implementation, mechanisms of change, and context [20,21].

The objective of this process evaluation was to assess implementation of the SPRING intervention using quantitative methods. The aim was to identify implications for the interpretation of trial outcomes, how well study findings might be reproduced and if any adaptations to the design should be made when implementing the intervention more widely or when designing similar trials in the future. Furthermore, we aimed to examine variations in effectiveness according to trial implementation [20].

Specifically, this paper addressed two main research questions:

1. How well were the SPRING interventions (HCS and vitamin D supplementation) implemented in terms of reach, fidelity, and dose?

2. How effective was the HCS intervention in improving diet quality and physical activity in subgroups defined according to the health behaviour (diet or physical activity) discussed?

## Methods

### The SPRING trial

The study protocol contains a detailed description of the SPRING trial and was published previously [14]. An overview of the study design is presented in S1 Fig. In brief, SPRING is a randomised controlled trial with a two-by-two factorial design. The two interventions under investigation were daily oral vitamin D supplementation and a behaviour change intervention (HCS). The study's primary aim was to improve pregnant women's nutritional status.

The study received ethical approval from the NRES Committee South Central Hampshire B (13/SC/0409) and was conducted according to the Declaration of Helsinki. All participants provided informed, written consent.

### Population and setting

The trial was undertaken in Southampton, UK and recruitment took place from 16 April 2014–03 April 2020. Southampton is a relatively deprived city on the south coast

of England and was ranked 55<sup>th</sup> most deprived out of 317 local authorities, according to the Index of Multiple Deprivation 2019 [22]. Participants were recruited opportunistically from the Princess Anne Hospital at around 12 weeks of gestation. Princess Anne Hospital is the main maternity hospital in Southampton and provides care for around 5,000 pregnant women each year. Women were recruited to the trial if they were aged over 18 years and had a singleton pregnancy of less than 17 weeks' gestation. Exclusion criteria, including in relation to metabolic or chronic diseases, are presented in the study protocol [14].

### Interventions

The vitamin D intervention required participants to ingest capsules containing 1,000 international units of cholecalciferol (vitamin D3) daily throughout the pregnancy. Participants assigned to the control group received a placebo capsule. Using a computer system, 1:1 randomisation was undertaken by the manufacturer before supplying the supplements to University Hospital Southampton's pharmacy, where they were collected by research nurses and research midwives. At the baseline appointment (14 weeks' gestation), the research nurses/ midwives issued the study medication (vitamin D/ placebo) in a box of blister-packed capsules to the participants. Women recruited after 14 weeks' gestation received the study medication at the earliest opportunity following recruitment. Both the participants and study personnel were blinded to the capsule's contents (vitamin D or placebo).

Healthy Conversation Skills (HCS) is a training intervention for practitioners to support their clients with behaviour change by increasing self-efficacy and empowerment. The training imparts four essential skills:

1. Asking "Open Discovery Questions", i.e., open-ended questions that generally begin with 'what' or 'how' and enable the participant or client to explore an issue, identify barriers, and generate solutions.

2. Listening more than talking, i.e., allowing time for the participant or client to find their own solutions instead of giving advice

3. Supporting SMARTER (Specific, Measurable, Action-oriented, Realistic, Timed, Evaluated, Reviewed) goal-setting

4. Reflecting on practice to achieve behaviour change and conversations

Block randomisation was used to assign participants to receive the behaviour change intervention or usual care. Three research nurses/ midwives trained in Healthy Conversation Skills had four Healthy Conversations with women assigned to the intervention group. Three of these were held face-to-face (at 14, 19, and 34 weeks' gestation) and one was over the phone (at 26 weeks' gestation). Participants recruited after 14 weeks' gestation had their first Healthy Conversation at the earliest opportunity following recruitment. Participants were not informed that the conversations formed part of the intervention. Nurses/ midwives who interacted with participants assigned to the control group did not receive HCS training. These research nurses/ midwives only carried out standard procedures including administration of study questionnaires and taking anthropometric measurements.

### Data collection

During pregnancy, participants completed two interviewer-led questionnaires, one at 14 weeks gestation and one at 34 weeks. These time points involved collecting data on participants' diet and physical activity behaviours, smoking status, alcohol consumption, self-efficacy, and perceived control. Ethnicity, educational attainment, and employment status were self-reported as part of the 14-week questionnaire. In the 34-week questionnaire, a modified version of the Problematic Experiences of Therapy Scale (PETS) was used to assess participants' perception of the extent to which they faced problems with taking the study medication [23]. A description of the measures used and how the data were treated can be found in S1 Table.

Pill counts were conducted at the 19- and 34-week appointments, and at delivery based on the blister packages brought to the visits to assess compliance with taking the study medication. Compliance was defined as the percentage of pills taken against the total number of pills expected to be taken by each woman [24]. Where available, the number of pills taken by the time of delivery was used to calculate compliance. If this information was missing, the latest count was used.

Anthropometric measurements including women's height and weight were taken by research nurses/ midwives at each appointment. Body-Mass-Index (BMI) categories were created using cut-off values as defined by the World Health Organisation (WHO) [25].

Healthy Conversations were recorded by the research nurses/ midwives using case report forms. Information on which health behaviour(s) were discussed (ranked for importance according to the detail in which they were discussed), the overall goal, and how long the Healthy Conversation lasted was recorded on these forms. Additionally, a random sample of the 26-week phone calls was audio-recorded to assess the research nurses'/ midwives' competence in using HCS.

## Evaluation of intervention implementation

Implementation of the SPRING interventions was assessed in accordance with the MRC guidance for the process evaluation of complex interventions [20]. Reach (whether the intended audience came into contact with the intervention), fidelity (whether the intervention was delivered as intended), and dose (the quantity of intervention implemented) were assessed.

**Reach.** Characteristics of study participants were analysed using descriptive statistics. To assess how well SPRING participants represented the population of pregnant women in England, a selection of indicators were compared with data published by Schoenaker et al. [26] and the Office for Health Improvement and Disparities (OHID) [27]. These data were taken from the Maternity Services Dataset (MSDS), a national source of patient-level data on pregnant women in England recorded as part of routine maternity care (version 1.5, 2018/19) [28]. Alcohol consumption was not included in this comparison because the indicators available in the MSDS were not comparable with the data collected in the SPRING trial. Dropout rates were analysed using descriptive statistics. Characteristics of participants who dropped out and those who stayed in the study were compared using statistical tests.

**Fidelity.** As the vitamin D supplement is an Investigational Medicinal Product (IMP), delivery of the intervention followed high quality processes with standard operating procedures (SOPs) followed, which were audited regularly by the study sponsor – University Hospital Southampton. Fidelity was therefore assumed to be high.

To assess fidelity of the HCS intervention, a random sample of 26-week phone calls was recorded and transcribed. The data were used to score each research nurses'/ midwives' competence in using three out of four Healthy Conversation Skills (asking Open Discovery Questions, listening, and supporting SMARTER goal-setting), using a coding rubric that was developed previously [29,30]. 'Reflecting on practice and conversations' was excluded from the competency scoring because it was not possible to assess this skill using audio recordings. For each skill, a rating of 0 (no evidence of competence) to 4 (high level of competence) was given. These were summed to obtain an overall HCS competency score for each Healthy Conversation held with participants. Scores ranged from 0 to 12. Responses were double-coded and discrepancies resolved through discussion. Data on the health behaviour(s) discussed were obtained from HCS case reports.

**Dose.** Compliance with taking the study medication was analysed separately for the groups who received vitamin D and placebo as well as both groups combined. To assess the problems participants had with taking the study medication, each sub-scale of PETS was dichotomised into "no problems" and "at least one problem" as described in S1 Table. Case report data were analysed using descriptive statistics to identify the number of Healthy Conversations had with each woman and the duration of these conversations.

## Efficacy of the HCS intervention in subgroups

Subgroups were defined according to the health behaviour participants mainly discussed – diet or physical activity, as recorded in the case report forms. Linear regression models were used to assess:

a)  The effect of the HCS intervention on diet quality in the subgroup of women who mainly discussed diet compared with participants in the control group,

b)  The effect of the HCS intervention on physical activity in the subgroup of women who mainly discussed physical activity compared with participants in the control group.

Diet quality was assessed using a 20-item Food Frequency Questionnaire (FFQ). The FFQ was developed to measure adherence to a prudent dietary pattern among women in Southampton [31]. The data obtained were used to create a standardised dietary quality score. The physical activity measure was created by calculating the average amount of time women spent being physically active per week. This total included gentle, moderate, and strenuous forms of physical activity. The physical activity data were highly skewed and therefore transformed to create a Fisher-Yates normal score. More details on both measures are provided in S1 Table.

Separate analyses were conducted regarding the health behaviours discussed at the 14- and 19-week appointment and during the 26-week phone call. The 34-week appointment was not included in this analysis because diet and physical activity outcomes were assessed at 34 weeks and any changes to these generated by the Healthy Conversation at 34 weeks would not have been captured. To assess whether the number of times the health behaviour was discussed (dose) had an impact on the outcome, the exposure variable was divided into four groups: i) participants in the control group, ii) those who discussed diet/physical activity as the main health behaviour once, iii) twice and iv) three times. Participants who were part of the intervention group but did not focus on these health behaviours were excluded from the analysis because they may have discussed diet/physical activity as part of a general conversation about health without setting primary goals to change diet and/or physical activity.

Confounders were identified by creating directed acyclic graphs (DAGs) [32,33] using the online software DAGitty [34]. Separate DAGs were created for the analyses investigating the effect of the HCS intervention on the dietary quality score and physical activity score in the respective subgroups (see S2 and S3 Figs). In accordance with the DAGs, age, educational attainment, level of neighbourhood deprivation as determined via Index of Multiple Deprivation (IMD) deciles [35], perceived control, and self-efficacy were added as covariables to the model for the diet subgroup. Age, self-efficacy, and weight were added to the model for the physical activity subgroup.

## Statistical analysis

Analyses were carried out using Stata version 17.0 (Statacorp, College Station, Texas, USA). Data are reported as mean ± standard deviation (SD) for continuous and normally distributed variables, median and interquartile range (IQR) for non-normally distributed variables, and absolute numbers and percentages (n, %) for categorical variables.

Differences between groups were assessed using an unpaired t-test for continuous and normally distributed variables, the Wilcoxon rank-sum test for continuous and non-normally distributed variables, and the chi-squared test for categorical variables with expected frequencies of greater than 5 in at least 80% of cells and greater than 1 in all cells. If requirements for the chi-squared test were not met, a two-sided Fisher's exact test was used.

For the subgroup analyses, multiple linear regression models were used to compare the intervention and control groups. To account for differences between the groups at baseline, analyses were adjusted for potential confounders.

The sample size for the SPRING trial was determined by the power calculation conducted for the effectiveness evaluation [14]. Complete case analyses were conducted. P-values are reported but no particular cut-off value was considered statistically significant [36]. Instead, results were interpreted with a focus on the effect size and range of values within the 95% confidence interval (CI).

Reporting followed the STROBE guidance (see S2 Table) [37]. Even though SPRING is a randomised controlled trial, a reporting guideline for observational studies was chosen because the data collected as part of the process evaluation were observational.

# Results

## Characteristics of study participants

The flow diagram of participants is shown in S4 Fig. A total of 717 women were recruited to the SPRING trial, randomised, and provided baseline data. Of these, 76 participants were recruited after the study medication could no longer be obtained. Therefore, these participants were only randomised to the HCS intervention or control and did not receive any supplements. In total, 366 women received the HCS intervention, while 351 were assigned to the control group. The vitamin D supplement was given to 321 participants and placebo to 320.

Characteristics of the SPRING participants are shown in Table 1. The average age was 31 years. Most women (94%) reported being of white ethnicity. The median level of deprivation was 6 and the IMD ranged from 1 (most deprived) to 10 (least deprived). More than half of the participants (51%) had high educational attainment (educated to degree level). Around a third (34%) of women were classified as having overweight and 22% had obesity. Of all participant characteristics, only the dietary quality score at baseline was different when comparing the HCS intervention and control group. The control group's dietary quality score was on average 0.17 SD (95%-CI, 0.03; 0.32) higher than that of the intervention group. The trial aimed to detect a difference of 0.25 SD in dietary quality score at 34 weeks' gestation between intervention and control participants. Further discussion of this difference at baseline is provided in the main outcome paper of the trial (in preparation).

**Table 1. Baseline characteristics (around 14 weeks gestation) of study participants in each arm of each intervention.**

| | | Healthy Conversations | | Vitamin D | |
|---|---|---|---|---|---|
| | | Yes n=366 | No n=351 | Yes n=321 | No n=320 |
| **Age [years]**, mean±SD | | 31.4±5.3 | 31.3±4.9 | 31.4±5.1 | 31.5±5.2 |
| **Ethnicity**, n % | White | 340 (93.2) | 332 (94.6) | 302 (94.1) | 301 (94.4) |
| | Black | 5 (1.4) | 4 (1.1) | 5 (1.6) | 3 (0.9) |
| | Asian | 14 (3.8) | 6 (1.7) | 7 (2.2) | 8 (2.5) |
| | Other | 6 (1.6) | 9 (2.6) | 7 (2.2) | 7 (2.2) |
| **Deprivation**, median (IQR) | Index of Multiple Deprivation | 6 (4; 8) | 6 (4; 8) | 6 (4; 8) | 6 (4; 8) |
| **Educational attainment**, n (%) | Low (None, CSE, O levels) | 63 (17.4) | 48 (13.7) | 52 (16.4) | 47 (14.7) |
| | Medium (A levels, HND) | 115 (31.8) | 122 (34.9) | 103 (32.5) | 104 (32.6) |
| | High (Degree) | 184 (50.8) | 180 (51.4) | 162 (51.1) | 168 (52.7) |
| **Number of children**, n (%) | 0 | 152 (41.8) | 141 (40.8) | 131 (41.3) | 128 (40.4) |
| | 1 | 142 (39.0) | 145 (41.9) | 126 (39.8) | 134 (42.3) |
| | 2 | 52 (14.3) | 45 (13.0) | 41 (12.9) | 43 (13.6) |
| | 3 or more | 18 (5.0) | 15 (4.3) | 19 (6.0) | 12 (3.8) |
| **Weight**, n (%) | Underweight | 4 (1.1) | 9 (2.6) | 3 (1.0) | 9 (2.8) |
| | Normal weight | 142 (39.0) | 154 (44.5) | 142 (44.9) | 123 (38.6) |
| | Overweight | 123 (33.8) | 119 (34.4) | 98 (31.0) | 113 (35.4) |
| | Obesity | 95 (26.1) | 64 (18.5) | 73 (23.1) | 74 (23.2) |
| **Diet Quality†**, mean±SD | Dietary Quality score | −0.09±0.97 | 0.09±1.03 | 0.02±0.97 | −0.00±1.02 |
| **Physical Activity‡**, median (IQR) | Hours spent being physically active per week | 2.0 (1.3; 2.8) | 2.0 (1.3; 3.0) | 2.0 (1.5; 3.0) | 2.0 (1.3; 3.0) |

This table presents each woman twice; once in the first two columns, indicating whether she received Healthy Conversations or not, and once in the second two columns, indicating whether she received the vitamin D supplementation or not. The numbers randomised to Healthy Conversations exceed those randomised to vitamin D due to unavailability of the vitamin D capsules at a late point in the trial. † Data missing for n=5. ‡ Data missing for n=15. CSE, Certificate of Secondary Education; HCS, Healthy Conversation Skills; HND, Higher National Diploma; IQR, Interquartile Range; SD, standard deviation.

## Reach

Compared with data on pregnant women in England, the proportion of women with advanced maternal age (≥ 35 years) was slightly higher among the SPRING sample, and the proportion of women under 20 years of age was slightly lower (Table 2). Women from ethnic minorities, as well as those living in the most deprived areas, were underrepresented. Furthermore, the proportion of women with overweight was higher among SPRING participants. Smoking at the baseline appointment or at the time of the maternity booking appointment was less prevalent among study participants than it was among pregnant women in England.

A total of 57 participants were lost to follow-up, corresponding to an overall dropout rate of 7.9%. The dropout rates did not differ between groups (p = 0.8). The most important reasons for dropping out of the study were clinical complications (31%), the inability or unwillingness to take the study medication (19%), and being too busy (9%).

Table 3 displays the differences between participants who stayed in the study and those who were lost to follow-up. Women living in the most deprived areas of England, who had lower educational attainment, who were unemployed during the last year, and who had multiple children were more likely to drop out. A higher percentage of women who were smoking at baseline withdrew from the study. There was also a tendency for women who drank more alcohol during the last three months and those with obesity to be more likely to drop out. The mean dietary quality score was 0.19 SDs (95%-CI, −0.09; 0.47) lower among dropouts compared with participants who did not withdraw.

## Vitamin D

**Fidelity.** SOPs and training of research nurses/ midwives ensured that the study medication was dispensed as planned. A small number of women (n = 76) were only randomised to receive the HCS intervention or control due to unavailability of the study medication. These participants received special identification numbers to make them easily identifiable and SOPs were amended accordingly.

Six women received supplements with an expiry date before their due dates. The doses missed due to short expiry period were recorded and ranged from 0 to 14.

**Dose.** With a median of around 96% of the vitamin D and placebo supplements taken, compliance with the intervention was high and did not differ between groups (Table 4). Around three quarters (76%) of women indicated that they did not have any problems taking the study medication. Uncertainties and doubts were seldom experienced, whereas practical problems were more common with 23% of women experiencing at least one practical problem. The most frequently

**Table 2. Comparison between study participants and pregnant women in England.**

| Indicator | SPRING participants at baseline (~ 14 weeks gestation) | | Pregnant women in England* (~ 9 weeks gestation) | |
|---|---|---|---|---|
| **Maternal age [years]**, mean ± SD | | 31.3 ± 5.1 | | 30.0 ± 5.7 |
| **Advanced maternal age**, % | Age ≥ 35 years | 24.7 | Age ≥ 35 years | 21.4 |
| **Teenage pregnancy**, % | Age < 20 years | 2.1 | Age < 20 years | 3.8 |
| **Ethnicity**, % | Other than White | 6.2 | Ethnic minorities | 22.8 |
| **Living in most deprived area (bottom 10%)**, % | Index of Multiple Deprivation Decile = 1 | 7.1 | Index of Multiple Deprivation Decile = 1 | 14.2 |
| **Employment status**, % | Unemployed in the last year | 9.5 | Unemployed and seeking work | 5.7 |
| **Weight**, % | Underweight | 1.8 | Underweight | 3.1 |
| | Overweight | 34.1 | Overweight | 28.0 |
| | Obesity | 22.4 | Obesity | 22.3 |
| **Tobacco use**, % | Currently smoking | 6.6 | Smoking in early pregnancy | 12.8 |

\* Data were obtained from Schoenaker et al. [26], except for 'Smoking in early pregnancy', which was obtained from the OHID Public Health Profiles (year 2018/2019) [27]. SD, standard deviation.

**Table 3. Characteristics of participants who dropped out and those who stayed in the study.**

| | | Dropouts n = 57 | Participants who stayed in the study n = 660 | P value |
|---|---|---|---|---|
| **Maternal age [years],** mean ± SD | | 30.9 ± 5.5 | 31.3 ± 5.1 | 0.5 |
| **Advanced maternal age,** n (%) | Age ≥ 35 years | 14 (24.6) | 163 (24.7) | >0.9 |
| **Teenage pregnancy,** n (%) | Age < 20 years | 1 (1.8) | 14 (2.1) | >0.9 |
| **Ethnicity,** n (%) | White | 54 (96.4) | 618 (93.6) | 0.60 |
| | Other | 2 (3.6) | 42 (6.4) | |
| **Deprivation,** median (IQR) | Index of Multiple Deprivation | 5 (2; 8) | 6 (4; 8) | 0.07 |
| **Living in most deprived area,** n (%) | Bottom 10% | 9 (16.4) | 41 (6.3) | 0.01 |
| **Educational attainment,** n (%) | Low (None, CSE, O levels) | 19 (35.2) | 92 (14.0) | < 0.001 |
| | Medium (A levels, HND) | 19 (35.2) | 218 (33.1) | |
| | High (Degree) | 16 (29.6) | 348 (52.9) | |
| **Employment status,** n (%) | Unemployed in the last year | 13 (23.2) | 55 (8.3) | < 0.001 |
| **Number of children,** n (%) | 0 | 14 (25.0) | 279 (42.7) | 0.002 |
| | 1 | 23 (41.1) | 264 (40.4) | |
| | 2 | 12 (21.4) | 85 (13.0) | |
| | 3 or more | 7 (12.5) | 26 (4.0) | |
| **Weight,** n (%) | Underweight | 2 (3.5) | 11 (1.7) | 0.09 |
| | Normal weight | 18 (31.6) | 278 (42.6) | |
| | Overweight | 18 (31.6) | 224 (34.3) | |
| | Obesity | 19 (33.3) | 140 (21.4) | |
| **Diet quality,** mean ± SD | Dietary quality score | − 0.18 ± 0.94 | 0.01 ± 1.00 | 0.18 |
| **Physical activity,** median (IQR) | Hours spent being physically active per week | 2.0 (1.3; 3.0) | 2.0 (1.3; 3.0) | 0.8 |
| **Tobacco use,** n (%) | Currently smoking | 11 (19.6) | 36 (5.5) | < 0.001 |
| **Alcohol consumption,** median (IQR) | Units of alcohol per week (over the last 3 months) | 1.8 (1.0; 3.0) | 1.2 (0.6; 2.0) | 0.06 |

CSE, Certificate of Secondary Education; HCS, Healthy Conversation Skills; HND, Higher National Diploma; IQR, Interquartile Range; SD, standard deviation.

**Table 4. Compliance and problems with taking the study medication.**

| | Vitamin D | Placebo | P value |
|---|---|---|---|
| **Compliance,** median (IQR) | 95.5 (89.3; 98.8) | 95.9 (88.4; 99.3) | > 0.9 |
| **At least one uncertainty,** n (%) | 1 (0.3) | 1 (0.3) | > 0.9 |
| **At least one doubt,** n (%) | 2 (0.7) | 1 (0.3) | 0.6 |
| **At least one practical problem,** n (%) | 62 (21.2) | 73 (24.3) | 0.4 |

IQR, Interquartile Range.

mentioned problem was difficulty remembering to take the capsules, as reported by 86% of those who indicated any practical problems.

## Healthy Conversation Skills

**Fidelity.** A total of 291 26-week phone calls were audio-recorded. Competency scoring showed that the trained research nurses/ midwives were highly proficient in using HCS (see Table 5 and S5 Fig). A high level of competence

**Table 5. Median (IQR) competency scores for three Healthy Conversation Skills.**

|  | Competency Score |
|---|---|
| **Asking Open Discovery Questions** | 4 (3; 4) |
| **Listening more than talking** | 4 (3; 4) |
| **SMARTER goal setting** | 3 (3; 4) |
| **Total** | 11 (10; 12) |

IQR, Interquartile Range.

(score of three to four) was demonstrated by research staff in asking open discovery questions and listening more than talking in 98% and 94% of the phone calls, respectively. A high level of competence in supporting SMARTER goal setting was evident in 86% of the Healthy Conversations.

Over the four different time points, between 65% and 75% of women talked about physical activity as the main health behaviour they wanted to change, while between 12% and 22% focused on their diets during the Healthy Conversations (Table 6). Among women who discussed multiple health behaviours, between 37% and 56% chose to talk about their diets as a second priority for change (see S3 Table).

**Dose.** Attendance was high at each of the four appointments where Healthy Conversations were held with a research nurse/ midwife (Table 7). Most women (85%) attended all four appointments, while 9%, 4%, and 2% attended three, two, and one, respectively. At each appointment/ phone call, participants were exposed to HCS for a median of 6–7 minutes.

### Efficacy of the HCS intervention in subgroups

As can be seen in S4 Table, our results indicated a slightly lower dietary quality score in the subgroups of women who mainly discussed diet at the 14-week appointment ($\beta = -0.06$ (95%-CI, −0.23; 0.12), n = 80), the 19-week appointment ($\beta = -0.02$ (95%-CI, −0.23; 0.19), n = 55) and during the 26-week phone call ($\beta = -0.10$ (95%-CI, −0.30; 0.10), n = 59) compared with control participants. However, the confidence intervals were wide and these results need to be interpreted with

**Table 6. Main health behaviours discussed at each of the four Healthy Conversations (n, %).**

|  | 14 weeks | 19 weeks | 26 weeks | 34 weeks |
|---|---|---|---|---|
| **Diet** | 80 (22.0) | 55 (15.3) | 59 (17.9) | 39 (11.6) |
| **Physical activity** | 236 (64.8) | 262 (72.8) | 247 (74.8) | 222 (66.1) |
| **Smoking** | 26 (7.1) | 15 (4.2) | 10 (3.0) | 14 (4.2) |
| **Alcohol** | 0 | 0 | 0 | 0 |
| **Breastfeeding** | 1 (0.3) | 11 (3.1) | 4 (1.2) | 45 (13.4) |
| **Study Medication** | 12 (3.3) | 10 (2.8) | 2 (0.6) | 4 (1.2) |
| **Other** | 9 (2.5) | 7 (1.9) | 8 (2.4) | 12 (3.6) |

**Table 7. Percentage of appointments where Healthy Conversations took place and median duration of the conversations at each appointment.**

| Appointment/ Phone call | Attendance, % | Duration [minutes], median (IQR) |
|---|---|---|
| Week 14 | 99.4 | 6 (5; 8) |
| Week 19 | 97.5 | 6 (5; 8) |
| Week 26 | 90.7 | 7 (5; 9) |
| Week 34 | 93.8 | 7 (5; 8) |

IQR, Interquartile Range.

caution because they are based on only a small number of observations. Moreover, comparison of baseline characteristics revealed that participants who discussed diet as the main health behaviour differed from those in the control group (see S5 Table). Importantly, women who primarily discussed diet had a substantially lower dietary quality score at baseline ($-0.38 \pm 0.83$, n = 133) compared with control ($0.09 \pm 1.03$, n = 351) and the rest of the intervention group ($0.08 \pm 1.00$, n = 233).

The relationship between the number of times diet was discussed as the main health behaviour and the effect on the diet quality score was not assessed due to very few observations in each group; only 15 women focused on their diets during all three Healthy Conversations.

Table 8 shows the association between exposure to the HCS intervention and physical activity at 34 weeks of gestation. The effect estimates for each subgroup are similar and suggest a higher level of physical activity (total of gentle, moderate, and strenuous intensity) in the intervention group compared with control at 34 weeks' gestation. The difference in level of physical activity was greatest in the subgroup of intervention women who primarily discussed physical activity during the 26-week phone call as compared with the control non-HCS subgroup ($\beta = 0.15$ ($-0.01$; $0.31$)). Our findings may indicate that the effect of the Healthy Conversations intervention was dependent on dose of HCS ($p_{trend} = 0.10$). Women who discussed physical activity as the main health behaviour across all three Healthy Conversations demonstrated greater levels of physical activity compared with controls (0.16 SD ($-0.03$; $0.34$)). This difference equates to an additional five minutes of physical activity per week for someone who takes the median amount of physical activity at baseline (2 hours per week).

Although there is no marked difference in baseline characteristics between women in the control group and those who discussed physical activity as the main health behaviour (see S6 Table), it is important to note that the participants who

**Table 8. Association between exposure to the Healthy Conversation Skills intervention and physical activity at 34 weeks of gestation in subgroups of women who mainly discussed physical activity.**

| | Adjusted* | | Crude† | |
|---|---|---|---|---|
| | β (95%-CI) | p value | β (95%-CI) | p value |
| **14-week appointment** | | | | |
| **Control** | ref | | Ref | |
| **Intervention** (n = 215) | 0.10 (−0.07; 0.27) | 0.2 | 0.08 (−0.08; 0.25) | 0.3 |
| | **Adjusted R²=7.6%** | | **Adjusted R²=6.9%** | |
| **19-week appointment** | | | | |
| **Control** | ref | | ref | |
| **Intervention** (n = 242) | 0.11 (−0.05; 0.28) | 0.2 | 0.08 (−0.08; 0.24) | 0.3 |
| | **Adjusted R²=8.1%** | | **Adjusted R²=7.5%** | |
| **26-week phone call** | | | | |
| **Control** | ref | | ref | |
| **Intervention** (n = 234) | 0.15 (−0.01; 0.31) | 0.06 | 0.12 (−0.04; 0.28) | 0.1 |
| | **Adjusted R²=8.7%** | | **Adjusted R²=7.6%** | |
| **Number of Healthy Conversations during which physical activity was discussed as the main health behaviour** | | | | |
| **Control** | ref | | ref | |
| **1** (n = 51) | 0.00 (−0.31; 0.32) | >0.9 | 0.04 (−0.27; 0.35) | 0.8 |
| **2** (n = 101) | 0.08 (−0.14; 0.31) | 0.5 | 0.06 (−0.16; 0.29) | 0.6 |
| **3** (n = 164) | 0.16 (−0.03; 0.34) | 0.1 | 0.12 (−0.07; 0.30) | 0.2 |
| | **Adjusted R²=7.2%** | | **Adjusted R²=6.1%** | |

* Adjusted for physical activity at baseline (standardised), age, Body-Mass-Index, and self-efficacy. † Adjusted for physical activity at baseline (standardised). CI, confidence interval.

discussed physical activity never or only once are very different from those who discussed it during all three Healthy Conversations (see S7 Table). Notably, among those who never discussed physical activity as the primary health behaviour, there was a higher percentage of women with low educational attainment (40.0% vs. 12.2% among those who discussed physical activity three times) and obesity (40.0% vs. 22.6%). Their mean dietary quality score was also considerably lower (−0.65 vs. 0.24).

## Discussion

This process evaluation shows that the SPRING trial achieved high rates of retention and compliance with both intervention components – Vitamin D supplementation and the HCS behaviour change programme. The small number of participants who were lost to follow-up differed from those who completed the trial. Women of lower socioeconomic status and those with fewer healthy behaviours were more likely to withdraw from the study. Women from ethnic minority communities and of lower socio-economic status were underrepresented in the SPRING trial; the increased likelihood of these women withdrawing from the trial adding to their underrepresentation.

The quality of delivery of both the vitamin D and the HCS intervention was high. During the Healthy Conversations, most women talked about physical activity as the primary health behaviour to improve and a smaller number of women focused on improving their dietary quality. Our findings suggest that diets are a more challenging topic on which to engage pregnant women and the HCS intervention did not show an improvement on dietary quality among women who primarily discussed diet. Among those who focused on improving their physical activity levels, a small increase in levels of physical activity was observed. There was some suggestion that this effect may be dependent on dose of HCS received.

### Reach

Our findings indicate that women of higher socioeconomic status and those with healthier dietary, physical activity and smoking behaviours were more likely to take part and remain in the study. This suggests that those who were more likely to benefit from the intervention were less likely to be recruited and more likely to be lost to follow-up. While a contributing factor might be the fact that women presenting for their first antenatal care appointment after 17 weeks' gestation were missed, these findings most likely reflect the challenges of recruitment in these groups.

The underrepresentation of ethnic minority and socioeconomically deprived groups is a widespread problem in clinical trials [38]. At the same time, women from ethnic minorities and deprived backgrounds are disproportionally affected by risk factors such as obesity, diabetes, and cardiovascular disease, which are already evident in the preconception period [26]. Interventions that aim to improve health behaviours have the potential to widen or narrow these health inequalities [39,40]. It, therefore, is important that a sufficient number of women from different ethnic and socioeconomic backgrounds are recruited to clinical trials to enable differences in intervention impacts to be assessed.

Potential strategies to improve recruitment rates include creating trial conditions that build women's trust and self-efficacy [41], offering greater flexibility about when and where women are seen, and using a person-centred approach that is sensitive to each woman's culture, daily context and background [42]. The latter could, for example, be achieved by enhancing the cultural competency of the research team, including people with the respective ethnic origins in the trial design or conducting outreach appointments in settings where these women feel comfortable [42]. To date, there remains a paucity of research on which methods work best.

### Fidelity

Delivery of the vitamin D intervention was carefully regulated and study procedures ensured that every participant received the right supplement. The delivery of the HCS intervention was also of high quality, with clear evidence of trained research nurses/ midwives asking open discovery questions, listening more than talking, and supporting SMARTER goal-setting. This finding is consistent with previous studies of the implementation of HCS which showed that the intervention training

programme produced health and social care professionals who asked more open discovery questions and spent less time making suggestions or giving information [29,30]. These studies did not find the same level of support for SMARTER goal-setting by HCS trained health and social care practitioners as was found in the study reported in this manuscript [30,43]. This finding suggests that clinical training of nurses and midwives may provide additional background skills that enable these professionals to adequately support goal setting as an evidence-based mechanism for facilitating behaviour change.

The health behaviour(s) women discussed during the Healthy Conversations were not predetermined and instead emerged from the conversation and the topics that participating women themselves were interested in discussing. Only a small number of women talked about diet as the main health behaviour during each Healthy Conversation. As diet quality was defined as the primary outcome for evaluating the effectiveness of the HCS intervention, this is likely to affect the ability to detect an effect of the intervention.

A key reason why women may find changing their dietary behaviour difficult is the unhealthy food environments to which women are exposed in almost every aspect of their lives [44,45]. The abundance, cheaper cost and appeal of unhealthy foods makes changing dietary behaviours difficult for many people, including pregnant women. An additional reason why women might not have wanted to discuss improving their dietary behaviour could be pregnancy-specific barriers to healthy eating such as nausea [10]. Data from the Southampton Women's Survey (SWS) revealed that approximately 89% of women suffered from nausea during early pregnancy [46]. To ease symptoms, women tend to adapt the way they eat. For example, among SWS participants, increasing severity of nausea was associated with a higher consumption of white bread and soft drinks [46]. Therefore, for some women improving their diet quality during pregnancy might not be a priority or feel achievable. Qualitative research conducted as part of the SPRING trial suggested that women also face barriers to being physically active during pregnancy (e.g., pain). However, these barriers to engaging in physical activity were reported less often compared with barriers to eating a healthy diet [10]. This difference may explain women's focus on physical activity improvement during their pregnancies and is an area for future research.

At the same time, it is important to note that women of low educational attainment and obese women were more likely not to discuss physical activity during the Healthy Conversations. For women of low SES, lower levels of health literacy and limited resources might make physical activity a low priority [47]. Women living with overweight and obesity might be concerned about exercising safely and also experience lack of time and opportunity to exercise [48]. In the future, these factors need to be addressed to ensure that the intervention reaches and resonates with all women, particularly those who may face greater barriers to engaging in physical activity during pregnancy.

## Dose

The median compliance with taking the study medication in SPRING was just as high as among participants who took part in MAVIDOS [24]. In contrast, SPRING participants reported experiencing fewer practical problems, uncertainties, and doubts. While 53% of women taking part in MAVIDOS reported experiencing at least one practical problem [24], only 23% of women in the SPRING sample reported the same.

The high number of women who engaged in all four Healthy Conversations indicates that routine maternity care is a good setting for implementing HCS because it provides the opportunity to expose women to the intervention regularly and works well in achieving high compliance rates. Research nurses/ midwives used HCS for a median of 6–7 minutes at each appointment. This short time frame highlights the potential for HCS to be more integrated into routine clinical practice because the time burden is not extensive. Extraction of this time from recordings also demonstrated that these conversations were often held while the research practitioners were completing other tasks such as anthropometrical measurements or taking bloods, which further illustrates the potential for its incorporation into routine practice [49].

## Efficacy of the HCS intervention in subgroups

Drawing a conclusion on the effect of HCS on dietary quality among women who primarily discussed diet was not possible because the results lacked precision and were based on a small number of observations. Furthermore, the markedly low dietary quality score at baseline among participants in the intervention group who were included in the subgroup analysis is likely to explain why lower dietary quality scores were observed among the intervention participants compared with control participants. A number of participants also raised diet in relation to feelings of nausea and this context is unlikely to lead to an improvement in dietary quality, further limiting the trial's ability to detect a beneficial effect of the intervention on dietary quality.

Two previous studies that have investigated the effect of HCS on diet quality and physical activity showed mixed results. A non-randomised controlled before and after evaluation [50] assessed the efficacy of HCS in improving dietary quality and physical activity among women from disadvantaged backgrounds but did not show a difference in the primary outcomes compared with control. However, the intervention had a protective effect on the intermediate outcomes self-efficacy and sense of control. The intervention's effect on sense of control was dependent on exposure. Consistent with our findings, this result indicates that those who are having regular exposure to Healthy Conversations might benefit from the intervention the most. Adam et al. [51] conducted a pilot randomised controlled trial evaluating the effect of HCS on diet quality and physical activity as secondary outcomes. The participants were pregnant women and the intervention was delivered by a registered dietitian. Study findings suggested a positive effect of the intervention on diet quality and sedentary behaviour. However, no difference in total physical activity was reported. In contrast, the findings presented here suggest that HCS may support small increases in total physical activity.

## Strengths and limitations

The present study provides a comprehensive assessment of the factors involved in the implementation of the SPRING interventions and followed MRC guidance. The retention rate was high, with only around 8% of participants lost to follow-up. Compliance rates were also high, indicating a high degree of motivation among trial participants. The trial was a slight adaptation from usual care but demonstrates coherence with usual practice, although women who opt to participate in a clinical trial may not be representative of the general population and this factor could have implications for generalisability of the study findings.

The analyses exploring the efficacy of the HCS intervention among participant subgroups likely lacked adequate statistical power to detect intervention effects among those who selected to discuss diet. The subgroups were defined according to the main health behaviour discussed, however, many women set goals for changing several health behaviours. The time spent discussing each health behaviour may not have been equal nor reflected the priority given by participants to implement changes. Furthermore, while SMARTER goal-setting was assessed, it was not assessed for each participants' health behaviour goal at each time-point. Such detailed assessment in future studies may provide more nuanced insight into specific mechanisms of action.

Physical activity was not *a priori* defined as a primary or secondary outcome of the trial. Furthermore, the physical activity measure used combined different intensities of physical activity. Thus, we were not able to capture if a woman, for example, decreased the amount of time she spent doing strenuous exercise and took up gentle exercise instead. However, the incorporation of assessments of gentle forms of physical activity facilitated the detection of overall differences in the time women spent being physically active, including low-intensity activities, which can offer important clinical outcomes when taken up by pregnant women [52].

Although the data were collected as part of a randomised controlled trial, the process evaluation analyses presented here were observational. To reduce bias in these process evaluation analyses, statistical models were adjusted for potential confounders, identified using DAGs [32,33].

## Implications for policy and practice

In the United Kingdom, the 'Making Every Contact Count' (MECC) program was introduced by Public Health England in 2016 with the aim of improving health behaviours to help prevent non-communicable diseases [53]. Using day-to-day interactions, anyone working with the public is encouraged to engage individuals in conversations about their health and wellbeing [53]. Staff are trained to deliver (very) brief behaviour change interventions, which include HCS [54]. Currently, the approach to delivering MECC varies across England leaving potential to scale up the use of HCS. Findings of this process evaluation shed a light on factors that need to be considered when planning to implement the intervention more widely.

Our analysis showed that routine maternity care is a suitable setting for implementing HCS in terms of achieving high participant compliance rates. Previous research showed that barriers such as lack of time and prioritisation can have a negative impact on the delivery of behaviour change interventions by healthcare practitioners [55,56]. These barriers need to be considered when implementing the intervention in everyday practice to avoid low levels of exposure – a critical factor for intervention success. Pregnancy-specific barriers may also pose a hurdle to implementing behaviour change, particularly in achieving moderate to high intensity physical activity or better dietary quality and could be specifically targeted in HCS training sessions with midwives.

Even though findings from this process evaluation suggest that diet change is a more challenging topic to raise with pregnant women, HCS as an intervention allowed participants to engage in a personalised way that is tailored towards their individual circumstances. In 2016, authors of the National Maternity Review's report "Better Births" shared their vision of a more personalised approach to maternity care that is centred around women's needs and decisions [57]. HCS training for maternity staff could play a part in this transformation.

This process evaluation, identified factors that might limit the interventions potent mechanism of action. These need to be considered when interpreting findings from the effectiveness evaluation and designing policy recommendations. Most importantly, the small number of women who focused on discussing diet indicates that dietary changes are more challenging to raise and address than physical activity, likely due to current everyday food environments not supporting healthy choices as the obvious choice [44,58]. Few participants set goals to improve their diet quality which may indicate the burden individuals feel when trying to think about changing their diets. There is evidence that citizens and parents are increasingly supportive of governments introducing food policies and regulations to restrict widespread promotion and availability of unhealthy foods and reshape food environments to be more reflective of current dietary recommendations [59,60].

## Recommendations for future research

HCS is person-centred and participants focus on changing those behaviours that are most relevant to them and their circumstances. As such, more specific behavioural targets may need to be assessed in future studies to capture, for example, changes in specific sedentary behaviours or certain food groups such as confectionary or fruit and vegetables that individuals may target when setting SMARTER goals [61]. While dietary quality scores are useful for ranking populations in terms of overall adherence to dietary recommendations, they may not be specific enough to detect changes at an individual level. Subgroup analyses can also help determine the effect of the intervention on a specific behaviour that was discussed or the subject of a goal for change. Trials should be powered accordingly. However, it should be noted that participants who discuss a certain health behaviour are likely to differ from those who do not. These differences can introduce a selection bias and should therefore be assessed and adjusted for accordingly.

Future research could explore in greater detail the factors underlying the preference for women to discuss changing their physical activity behaviours over dietary behaviours during Healthy Conversations. These insights would help to shape improvements to the intervention, particularly because participants in this trial were generally motivated and engaged, as indicated by the high compliance rates. Furthermore, future research would benefit from understanding the

reasons why women who raised diet as the main health behaviour for change had poorer quality diets at baseline compared with those in the intervention group who did not talk about diet as the main health behaviour. In contrast, women who engaged in conversations about physical activity had a higher diet quality and socio-economic status than those who did not.

## Conclusions

This process evaluation provides insight into factors to consider when interpreting findings of the outcome evaluation and when planning implementation of behaviour change and supplementation interventions among pregnant women more generally. The quality of intervention delivery was high. Pregnant women tended to opt for making behaviour change goals to improve their physical activity levels rather than improve their dietary quality. This finding may indicate the burden individuals feel when trying to change their diets in mostly unhealthy food environments but may also reflect the inherent challenges of making dietary changes during pregnancy. Further research is warranted to understand how best to support dietary change through pregnancy. When evaluating person-centred behaviour change interventions, such as HCS, it is important to assess which behaviours the participants discussed and how this might affect trial outcomes. Future trials also need to adopt strategies that enable effective recruitment of women from ethnic minorities and deprived backgrounds.

## Supporting information

**S1 Fig. Overview of the study design.**
(DOCX)

**S2 Fig. Directed acyclic graph used to inform the model estimating the total effect of the Healthy Conversation Skills (HCS) intervention on the dietary quality score among women who discussed diet as the main health behaviour.**
(DOCX)

**S3 Fig. Directed acyclic graph used to inform the model estimating the total effect of the Healthy Conversation Skills (HCS) intervention on physical activity among women who discussed physical activity as the main health behaviour.**
(DOCX)

**S4 Fig. Participant flow diagram.**
(DOCX)

**S5 Fig. Trained research nurses' competency scores in using three Healthy Conversation Skills at the 26-week phone call.**
(DOCX)

**S1 Table. Data collection and processing of participants' characteristics and health behaviours.**
(DOCX)

**S2 Table. STROBE Statement – checklist of items that should be included in reports of observational studies.**
(DOCX)

**S3 Table. Health behaviour discussed in the second most detail at each Healthy Conversation.**
(DOCX)

**S4 Table. Association between exposure to the Healthy Conversation Skills intervention and diet quality at 34 weeks of gestation in subgroups of women who mainly discussed diet.**
(DOCX)

**S5 Table. Baseline characteristics of the control group and participants included in and excluded from the diet subgroup.**
(DOCX)

**S6 Table. Baseline characteristics of the control group and participants included in and excluded from the physical activity subgroup.**
(DOCX)

**S7 Table. Baseline characteristics of participants in the control group and those in the intervention group who discussed physical activity as the main health behaviour never, once, twice and three times.**
(DOCX)

## Acknowledgments

We thank the women who took part in the SPRING trial and gave us their time. We would also like to thank Chloe Gwynne, Sarah Jenner and Tannaze Tinati for conducting the HCS competency scoring.

## Author contributions

**Conceptualization:** Simone Proebstl, Christina Vogel, Wendy Lawrence, Sofia Strömmer, Mary Barker, Janis Baird.

**Formal analysis:** Simone Proebstl.

**Funding acquisition:** Nicholas C. Harvey, Mary Barker, Janis Baird.

**Investigation:** Christina Vogel, Wendy Lawrence, Sofia Strömmer, Nicholas C. Harvey, Mary Barker, Janis Baird.

**Project administration:** Christina Vogel, Wendy Lawrence, Sofia Strömmer, Nicholas C. Harvey, Mary Barker, Janis Baird.

**Supervision:** Hazel Inskip, Mary Barker, Janis Baird.

**Visualization:** Simone Proebstl.

**Writing – original draft:** Simone Proebstl.

**Writing – review & editing:** Christina Vogel, Wendy Lawrence, Sofia Strömmer, Hazel Inskip, Julia Hammond, Kate Hart, Karen McGill, Nicholas C. Harvey, Mary Barker, Janis Baird.

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
