## [Decision Letter · Decision Letter 0]

10 Jun 2025

PONE-D-25-02483Process evaluation of a randomised controlled trial aimed at improving health behaviours and vitamin D status during pregnancy: Implementation of the SPRING trialPLOS ONE

Dear Dr. Baird,

Thank you for submitting your manuscript to PLOS ONE. After careful consideration, we feel that it has merit but does not fully meet PLOS ONE’s publication criteria as it currently stands. Therefore, we invite you to submit a revised version of the manuscript that addresses the points raised during the review process.

You should address all the queries raised by the reviewers.

We look forward to receiving your revised manuscript.

Kind regards,

Francis Xavier Kasujja

Academic Editor

PLOS ONE

Journal Requirements:

“This work was supported by the UK Medical Research Council [MC_PC_21003; MC_PC_21001] and National Institute for Health and Care Research National Institute for Health Research (NIHR) Southampton Biomedical Research Centre, University of Southampton and University Hospital Southampton NHS Foundation Trust. The work leading to these results was supported by the European Union’s Seventh Framework Programme (FP7/2007–2013), projects EarlyNutrition, ODIN and LifeCycle under grant agreements numbers 289346, 613977 and 733206, and by the BBSRC (HDHL-Biomarkers, BB/P028179/1 and BB/P028187/1), as part of the ALPHABET project, supported by an award made through the ERA-Net on Biomarkers for Nutrition and Health (ERA HDHL), Horizon 2020 grant agreement number 696295. We are extremely grateful to Merck GmbH for the kind provision of the Vigantoletten supplement. The funders had no role in study design, data collection and analysis, decision to publish, or preparation of the manuscript.

For the purpose of Open Access, the author has applied a Creative Commons Attribution (CC BY) licence to any Author Accepted Manuscript version arising from this submission.”

“Janis Baird, Mary Barker and Wendy Lawrence have received grant research support from Danone Nutricia Early Nutrition. Cyrus Cooper has received consultancy, lecture fees and honoraria from AMGEN, GSK, Alliance for Better Bone Health, MSD, Eli Lilly, Pfizer, Novartis, Servier, Merck, Medtronic and Roche. Members of Hazel Inskip’s team have received grant research support from Nestec and Danone Nutricia Early Life Nutrition. Nicholas Harvey has received consultancy, lecture fees and honoraria from Alliance for Better Bone Health, AMGEN, MSD, Eli Lilly, Servier, Shire, UCB, Consilient Healthcare, Theramex, Kyowa Kirin and Internis Pharma.”

We note that one or more of the authors are employed by a commercial company

5. In the online submission form, you indicated that your data is available only on request from a third party. Please note that your Data Availability Statement is currently missing [the name of the third party contact or institution / contact details for the third party, such as an email address or a link to where data requests can be made]. Please update your statement with the missing information.

Reviewers' comments:

Reviewer's Responses to Questions

**Comments to the Author**

1. Is the manuscript technically sound, and do the data support the conclusions?

Reviewer #1: Partly

Reviewer #2: Yes

2. Has the statistical analysis been performed appropriately and rigorously? 

Reviewer #1: No

Reviewer #2: Yes

3. Have the authors made all data underlying the findings in their manuscript fully available?

Reviewer #1: Yes

Reviewer #2: Yes

4. Is the manuscript presented in an intelligible fashion and written in standard English?

Reviewer #1: Yes

Reviewer #2: Yes

5. Review Comments to the Author

Reviewer #1: Differences between groups were assessed using normal univariate statistics for continuous data and non parametric for non normal data. For the subgroup analyses, multiple linear regression models were used to compare the intervention and control groups. The R square values on the multiple regression might be helpful. To account for differences between the groups at baseline, analyses were adjusted for potential confounders. The analysis as discussed in the statistical section was routine as one would expect in this type of application. The analysis was done appropriately but could have been improved. Also, the analysis was presented somewhat fragmented. There were many pieces to try to digest separately. Groups were compared but at various times. What about a group or time effect overall? Was there any group by interaction effect? Figure S3 raises issues as to what is possibly the best intervention compared to control?

Fidelity, dose, and reach were evaluated descriptively. It is not really clear how much this added to the entire objectives of the study which mainly was to compare the vitamin D to the Healthy Conversation groupings. The design of the study was of the factorial nature. Sample size calculations were implemented.

One tries to get an idea of the flow of the study in a graphic and Figure S3 was not very helpful. A good schematic of the mechanics of the study showing what happens and when would have been more helpful.

The conclusion appears to dwell on intervention delivery and the mechanics of such. Where exactly is there a statement that states the women’s nutritional status was improved or not overall?

Reviewer #2: The manuscript is clearly written and of high quality. The results are very valuable for contextualizing the study's efficacy findings and to understand the implementation, dose and reach of the intervention.

Introduction/Methods: Please provide brief background information on the Healthy Conversation Skills intervention, especially whether this intervention has been implemented and evaluated among pregnant women in other settings, and if so, what the general results were regarding its efficacy.

Intervention Line 131: Please clarify the context of Vitamin D supplementation among pregnant women in the study area. Are women in the study area advised by their health care providers to take their own Vitamin D supplementation or were they relying on it being given by the hospital? If the former, could some participants in both control and intervention groups have taken Vitamin D and could that have confounded the final trial's efficacy results?

Line 136: Participants were recruited at less than 17 weeks’ gestation. Participants recruited between 15-17 weeks would have missed their baseline appointment at 14 weeks. When did they receive their capsules?

Line 154: same question as above for the Healthy Conversations with women at 14 weeks and the baseline interview at 14 weeks.

Line 171: “Pill counts were conducted at the 19- and 34-week appointments..”: We assume pill counts were self-reported, please specify.

Results Line 320-322: Pregnant women were recruited from the main maternity hospital in Southhampton. The Reach results can be contextualised with respect to how representative the maternity hospital setting is of all pregnant women in Southhampton. Please clarify if some Southhampton women seek antenatal care at other types of health facilities other than this maternity hospital. This could contextualise why the SPRING sample included higher proportions of older women and lower proportions of younger women. Could it be that younger women access antenatal care from the maternity hospital at late stages in their pregnancy (beyond 17 weeks) or younger women and those from ethnic minorities access care from other types of facilities?

The focus of the Healthy Conversation was determined by what each woman wanted to discuss as the primary health behaviour (e.g. physical activity/diet/other). Women with low educational attainment and obese women were more likely to never discus physical activity as the primary health behaviour. This could be due to lack of awareness/knowledge/health literacy on the importance of physical activity or perceived behavioural control, or lack of motivation, among other reasons. While not the focus of the study, could the authors recommend or comment on ways in which this intervention, or future interventions, could promote women (particularly those with low educational attainment or who are obese) to start talking about ways to improve their physical activity during pregnancy and to recognise the importance of physical activity.

6. PLOS authors have the option to publish the peer review history of their article (what does this mean?). If published, this will include your full peer review and any attached files.

Reviewer #1: No

Reviewer #2: **Yes: **Ronel Sewpaul

---

## [Author Response · Author response to Decision Letter 1]

22 Jul 2025

Francis Xavier Kasujja

Academic Editor

PLOS ONE

21st July 2025

Dear Dr Kasujja,

Thank you for the opportunity to revise and resubmit our manuscript entitled “Process evaluation of a randomised controlled trial aimed at improving health behaviours and vitamin D status during pregnancy: Implementation of the SPRING trial”.

We would like to thank the editor and reviewers for their careful evaluation of our manuscript and their insightful feedback. We have carefully addressed all comments and revised our manuscript accordingly. We believe that this helped improve the quality of our work.

Below, we provide a detailed response to each of the comments. Where applicable, we have noted the specific changes made to the manuscript. We hope that the revisions satisfactorily address the concerns raised and that the manuscript is now suitable for publication in PLOS ONE.

Response to the Editor

Author’s reply: We have made sure that our manuscript adheres to PLOS ONE's style requirements.

2. Thank you for stating in your Funding Statement: […] For the purpose of Open Access, the author has applied a Creative Commons Attribution (CC BY) licence to any Author Accepted Manuscript version arising from this submission.”

Author’s reply: We have updated the Funding Statement accordingly. Please find the revised Funding Statement at the end of this section.

3. Thank you for stating the following in the Competing Interests section: […] We note that one or more of the authors are employed by a commercial company

Author’s reply: While Professors Harvey and Cooper have carried out consultancy work, as indicated in our previous statement, neither of them is employed by these companies. So, we are unclear why you have suggested they are employed by a commercial company. The facts we have stated are correct in relation to their competing interests. We have added that their consultancy work did not play a role in this study. We trust that this addresses your comments.

Author’s reply: We have added the statement about PLOS ONE policies on sharing data and materials to our competing interests statement.

Author’s reply: We have reviewed our ethics and governance arrangements for the study and, as a result, we confirm that we will make our data available in the University of Southampton open access repository (PURE https://pure.soton.ac.uk/).

5. In the online submission form, you indicated that your data is available only on request from a third party. Please note that your Data Availability Statement is currently missing [the name of the third party contact or institution / contact details for the third party, such as an email address or a link to where data requests can be made]. Please update your statement with the missing information.

Author’s reply: Due to the updates made to our Data Availability Statement, this is no longer applicable.

Please find below the updated Funding and Competing Interests Statements.

Funding Statement

This work was supported by the UK Medical Research Council [MC_PC_21003; MC_PC_21001] and National Institute for Health and Care Research National Institute for Health Research (NIHR) Southampton Biomedical Research Centre, University of Southampton and University Hospital Southampton NHS Foundation Trust. The work leading to these results was supported by the European Union’s Seventh Framework Programme (FP7/2007–2013), projects EarlyNutrition, ODIN and LifeCycle under grant agreements numbers 289346, 613977 and 733206, and by the BBSRC (HDHL-Biomarkers, BB/P028179/1 and BB/P028187/1), as part of the ALPHABET project, supported by an award made through the ERA-Net on Biomarkers for Nutrition and Health (ERA HDHL), Horizon 2020 grant agreement number 696295. We are extremely grateful to Merck GmbH for the kind provision of the Vigantoletten supplement. The funders had no role in study design, data collection and analysis, decision to publish, or preparation of the manuscript. There was no additional external funding received for this study.

The funders (UK MRC, NIHR Southampton BRC, University of Southampton and University Hospital Southampton NHS Foundation Trust) provided support in the form of salaries for authors [NCH, JB, MEB, CC, MB, WTL, CV, SS, HMI, JH, KM, KH] but did not have any additional role in the study design, data collection and analysis, decision to publish, or preparation of the manuscript. The specific roles of these authors are articulated in the ‘author contributions’ section.

For the purpose of Open Access, the author has applied a Creative Commons Attribution (CC BY) licence to any Author Accepted Manuscript version arising from this submission.

Competing Interests Statement

Janis Baird, Mary Barker and Wendy Lawrence have received grant research support from Danone Nutricia Early Nutrition. Cyrus Cooper has received consultancy, lecture fees and honoraria from AMGEN, GSK, Alliance for Better Bone Health, MSD, Eli Lilly, Pfizer, Novartis, Servier, Merck, Medtronic and Roche. Members of Hazel Inskip’s team have received grant research support from Nestec and Danone Nutricia Early Life Nutrition. Nicholas Harvey has received consultancy, lecture fees and honoraria from Alliance for Better Bone Health, AMGEN, MSD, Eli Lilly, Servier, Shire, UCB, Consilient Healthcare, Theramex, Kyowa Kirin and Internis Pharma. The commercial companies that Professors Harvey and Cooper undertook consultancy work for did not play any role in this study. Professors Havey and Cooper were not directly employed by these companies. None of the interests declared alter our adherence to PLOS ONE policies on sharing data and materials.

Response to the Reviewers

Reviewer #1

Differences between groups were assessed using normal univariate statistics for continuous data and non parametric for non normal data. For the subgroup analyses, multiple linear regression models were used to compare the intervention and control groups. The R square values on the multiple regression might be helpful.

Author’s reply: We have added the R squared values to Tables 8 and S4 as requested. However, we feel that this makes the tables rather complicated to read and we will leave it to the editor’s discretion to decide if this change should be kept.

To account for differences between the groups at baseline, analyses were adjusted for potential confounders. The analysis as discussed in the statistical section was routine as one would expect in this type of application. The analysis was done appropriately but could have been improved. Also, the analysis was presented somewhat fragmented. There were many pieces to try to digest separately.

Author’s reply: We have reordered the results section to present our findings in a sequence that we hope improves clarity and coherence.

Groups were compared but at various times. What about a group or time effect overall? Was there any group by interaction effect?

Author’s reply: Our senior statistician has reviewed this and is not sure what is meant by a group by interaction effect. We believe that our analysis of data collected at the final 34-week time point, adjusted for the 14-week baseline measure, effectively takes account of group and time.

Figure S3 raises issues as to what is possibly the best intervention compared to control?

Author’s reply: Figure S3 is our CONSORT diagram. This diagram is designed to explain the flow of participants through the trial, not to compare intervention to control.

Fidelity, dose, and reach were evaluated descriptively. It is not really clear how much this added to the entire objectives of the study which mainly was to compare the vitamin D to the Healthy Conversation groupings. The design of the study was of the factorial nature. Sample size calculations were implemented.

Author’s reply: It was our aim to only investigate the interventions’ implementation - namely reach, fidelity, and dose - and to evaluate how variations in implementation influenced the effectiveness of the intervention. The trial outcomes will be published separately. We have added the following sentence to the Introduction section (ll. 92-93 in tracked changes version) for clarification:

The results of the outcome evaluation will be published separately and will be informed by the findings of this process evaluation.

One tries to get an idea of the flow of the study in a graphic and Figure S3 was not very helpful. A good schematic of the mechanics of the study showing what happens and when would have been more helpful.

Author’s reply: Figure S3 (now renamed to Figure S4) is a CONSORT diagram and adheres to the CONSORT statement. To address your comment, we added an overview of the study in the Supporting Information (S1 Fig) and added a reference to the additional figure in the Methods section under “The SPRING trial” (l. 113 in tracked changes version):

An overview of the study design is presented in S1 Fig.

The conclusion appears to dwell on intervention delivery and the mechanics of such. Where exactly is there a statement that states the women’s nutritional status was improved or not overall?

Author’s reply: As stated in our reply above, the focus of our manuscript is the evaluation of the intervention’s implementation, not an outcome assessment. We hope that the sentence added to the manuscript (see reply above) provides the necessary clarification for the reader.

Reviewer #2

The manuscript is clearly written and of high quality. The results are very valuable for contextualizing the study's efficacy findings and to understand the implementation, dose and reach of the intervention.

Introduction/Methods: Please provide brief background information on the Healthy Conversation Skills intervention, especially whether this intervention has been implemented and evaluated among pregnant women in other settings, and if so, what the general results were regarding its efficacy.

Author’s reply: As indicated in our response to Reviewer 1, this paper focuses on process and is not reporting outcomes. The outcome assessment will be published separately. Therefore, we have not added background information on the evaluation of the HCS intervention in the Introduction and Methods section. We have provided a brief overview of previous HCS assessments in the Discussion section (see ll. 583-598 in tracked changes version).

Intervention Line 131: Please clarify the context of Vitamin D supplementation among pregnant women in the study area. Are women in the study area advised by their health care providers to take their own Vitamin D supplementation or were they relying on it being given by the hospital? If the former, could some participants in both control and intervention groups have taken Vitamin D and could that have confounded the final trial's efficacy results?

Author’s reply: The trial assessed the effect of the prescription of Vitamin D delivered as part of the intervention. Some women in the study might have chosen to take Vitamin D themselves, consistent with public health advice, but our measurements at baseline showed little difference in Vitamin D levels between intervention and control.

Line 136: Participants were recruited at less than 17 weeks’ gestation. Participants recruited between 15-17 weeks would have missed their baseline appointment at 14 weeks. When did they receive their capsules?

Author’s reply: Participants who were recruited later than 14 weeks’ gestation were given their capsules as soon after recruitment as possible. We added a sentence to the Methods section under “Interventions” (ll. 141-142 in tracked changes version) to clar

---

## [Decision Letter · Decision Letter 1]

7 Aug 2025

Process evaluation of a randomised controlled trial aimed at improving health behaviours and vitamin D status during pregnancy: Implementation of the SPRING trial

PONE-D-25-02483R1

Dear Dr. Baird,

We’re pleased to inform you that your manuscript has been judged scientifically suitable for publication and will be formally accepted for publication once it meets all outstanding technical requirements.

Kind regards,

Francis Xavier Kasujja

Academic Editor

PLOS ONE

Additional Editor Comments (optional):

Reviewers' comments:

Reviewer's Responses to Questions

**Comments to the Author**

1. If the authors have adequately addressed your comments raised in a previous round of review and you feel that this manuscript is now acceptable for publication, you may indicate that here to bypass the “Comments to the Author” section, enter your conflict of interest statement in the “Confidential to Editor” section, and submit your "Accept" recommendation.

Reviewer #1: All comments have been addressed

2. Is the manuscript technically sound, and do the data support the conclusions?

Reviewer #1: (No Response)

3. Has the statistical analysis been performed appropriately and rigorously? 

Reviewer #1: (No Response)

4. Have the authors made all data underlying the findings in their manuscript fully available?

Reviewer #1: (No Response)

5. Is the manuscript presented in an intelligible fashion and written in standard English?

Reviewer #1: (No Response)

6. Review Comments to the Author

Reviewer #1: (No Response)

7. PLOS authors have the option to publish the peer review history of their article (what does this mean?). If published, this will include your full peer review and any attached files.

Reviewer #1: No

---

## [Editor Report · Acceptance letter]

PONE-D-25-02483R1

PLOS ONE

Dear Dr. Baird,

I'm pleased to inform you that your manuscript has been deemed suitable for publication in PLOS ONE. Congratulations! Your manuscript is now being handed over to our production team.

Kind regards,

on behalf of

Dr. Francis Xavier Kasujja

Academic Editor

PLOS ONE